# An Overview of Optical and Electrochemical Sensors and Biosensors for Analysis of Antioxidants in Food during the Last 5 Years

**DOI:** 10.3390/s21041176

**Published:** 2021-02-07

**Authors:** Maryam Nejadmansouri, Marjan Majdinasab, Gilvanda S. Nunes, Jean Louis Marty

**Affiliations:** 1Department of Food Science & Technology, School of Agriculture, Shiraz University, Shiraz 71441-65186, Iran; m.nejadmansouri@shirazu.ac.ir (M.N.); majdinasab@shirazu.ac.ir (M.M.); 2Pesticide Residue Analysis Center, Federal University of Maranhao, 65080-040 Sao Luis, Brazil; gilvanda.nunes@ufma.br; 3Faculty of Sciences, University of Perpignan Via Domitia, 52 Avenue Paul Alduy, 66860 Perpignan CEDEX 9, France

**Keywords:** antioxidant, phenolic compounds, sensors, biosensor, nanomaterial, enzyme, DNA, cell

## Abstract

Antioxidants are a group of healthy substances which are useful to human health because of their antihistaminic, anticancer, anti-inflammatory activity and inhibitory effect on the formation and the actions of reactive oxygen species. Generally, they are phenolic complexes present in plant-derived foods. Due to the valuable nutritional role of these mixtures, analysis and determining their amount in food is of particular importance. In recent years, many attempts have been made to supply uncomplicated, rapid, economical and user-friendly analytical approaches for the on-site detection and antioxidant capacity (AOC) determination of food antioxidants. In this regards, sensors and biosensors are regarded as favorable tools for antioxidant analysis because of their special features like high sensitivity, rapid detection time, ease of use, and ease of miniaturization. In this review, current five-year progresses in different types of optical and electrochemical sensors/biosensors for the analysis of antioxidants in foods are discussed and evaluated well. Moreover, advantages, limitations, and the potential for practical applications of each type of sensors/biosensors have been discussed. This review aims to prove how sensors/biosensors represent reliable alternatives to conventional methods for antioxidant analysis.

## 1. Introduction

In the recent ten years, the utilization of antioxidants has risen considerably in the food industry. Antioxidants are able to stop the oxidation of products throughout processing, storage, diffusion, and utilization, that is a main factor of guarantee food grade quality [1,2]. Antioxidants in foods can be classified into natural and synthetic. Natural antioxidants consist of the tocopherols (vitamin E), ascorbate (vitamin C), carotenoids, polyphenolic compounds such as flavonoids, phenolic acids, anthocyanins, proteins, and minerals; and synthetic antioxidants include butylated hydroxyanisole (BHA), propyl gallate (PG), *tert*-butylhydroquinone (TBHQ), and butylated hydroxytoluene (BHT), that are frequently used in food formulations [3,4,5]. Antioxidants chelate and scavenge free radicals, thus they avoid the damage caused by free radicals to the human body [6]. Antioxidants combat diseases that are derived from oxidative stress such as heart disease, cancer, diabetes, cardiovascular diseases, neurodegenerative disorders, AIDS, ageing, arthritis, asthma, autoimmune diseases, Alzheimer’s disease, Parkinson’s dementia, hypertension, cataracts, etc. [7,8,9,10,11]. Oxidative stress is a devastating procedure caused by high levels of reactive oxygen/nitrogen species (ROS/RNS) or depletion of the antioxidant matrix. This process induces a disproportion between the oxidative and antioxidant systems that can stimulate rapid cellular death [5,12]. In this sense, it is vital to recognize antioxidants in food because of health beneficial functions and lower danger of diseases attached to oxidative stress [13].

Several methods have been developed for identifying antioxidant capacity that differ in their procedures, complexities and applications [14]. Generally, antioxidant activity can be measured by instruments such as high performance liquid chromatography (HPLC), gas chromatography (GC), Fourier transform infrared spectroscopy (FT-IR), nuclear magnetic resonance (NMR), and capillary electrophoresis (CE) [15]. Also, several assays, such as oxygen radical absorbance capacity (ORAC), Folin-Ciocalteu (FC), 2,2′-azinobis-3-ethylbenzothiazoline-6-sulfonic acid)/Trolox equivalent (ABTS/TEAC; antioxidant capacity, 2,2-diphenyl-1-picrylhydrazyl (DPPH), ferric reducing antioxidant power (FRAP) and cupric reducing antioxidant capacity (CUPRAC) are used for antioxidant analysis [16,17]. These conventional techniques are costly, time consuming, require complicated procedures, various steps, overpriced reagents, particular apparatus to do the analysis, trained personnel to operate it and laborious sample pre-treatment processes. Therefore, the importance of new methods for antioxidant assessment based on sensors/biosensors is at the center of attention by the scientific groups and consumers, who are worried about health [6,14,18,19].

In recent times, optical and electrochemical sensors/biosensors have been used for antioxidant activity estimation due to their high reactivity and selectivity [12]. Sensors/biosensors are a reliable substitute for classical techniques. They are versatile because they are able to use as a supportive approach beside traditional ones, mainly when the basis of research is based on comprehension the character of each molecule in a matrix such as, plant sources that complex due to the presence of different classes of phenols. Also, sensors have the advantage of reducing the detection limit and expanding specificity and selectivity by using several nanomaterials or polymers, when the focus is on particular analytes [20]. Generally, sensors/biosensors have several advantages such as low price, flexibility, portability, ease of use, the possibility to use them at remote locations like the home, fast analysis time, speed, uncomplicated operation, robustness, reproducibility, long-term stability, minimal need for sample pretreatment, miniaturization, and on-site/in situ analysis [2]. Different studies have been reported by researchers for identifying antioxidant activity by optical and electrochemical sensors and biosensors in food. This review presents recent innovations in optical and electrochemical sensors and biosensors for the recognition and quantification of antioxidants in food published during the last five years (Scheme 1).

## 2. Optical Sensors and Assays

Optical sensors and assays investigate analytical data by utilizing optical transduction approaches such as, absorbance, reflectance, luminescence, etc. These techniques are proper for colored and turbid samples [12]. This review focuses on colorimetric and fluorescence assays which have been more widely used for antioxidant analysis than other optical techniques.

### 2.1. Colorimetric Sensors

Colorimetric assays are amongst the most common techniques for antioxidant activity analysis. They are especially well-suited for on-site sensing because of their simple readout and operation. The lightness of these tools is useful for industrial companies to verify their products’ conformity with regulatory limits. Colorimetric assays are classified into solution-based and solid substrate-based, based on the medium in which the reaction develops. Solid substrate-based methods have recently received much attention for antioxidant analysis. Solid substrates can be selected from a variety of materials, of which paper is one of the most popular. Paper-based devices are currently known as an effective and alternative method and have been applied as a detection platform in several areas, including food safety, environmental monitoring and clinical analysis [21]. Paper-based assays possess several advantages, they are simple, economical, portable, convenient to fabricate and use, non-returnable and require low sample and reagent consumption [15].

Different kinds of color generating probes are used in colorimetric assays such as dyes, enzymes and nanomaterials. Noble metal nanoparticles are the most frequent ones used in colorimetric assays for the detection of antioxidant capacity. They include gold nanoparticles (AuNPs), silver nanoparticles (AgNPs), cerium oxide nanoparticles (CeNPs, nanoceria), etc [17]. Nanoparticle-based colorimetric sensors have particular advantages such as tiny size, great particular surface area, high reactivity, sensitivity, stability, selectivity, and the possibility of implementing the tests on paper and other patterns that can be matched to optical apparatus suitable for automated investigation [18,22].

#### 2.1.1. AuNPs-Based Colorimetric Assays

AuNPs are extensively used in colorimetric assays due to their easy and cost-effective synthesis and simplicity of use. Basically, the principle of AuNPs-based assays’ response is a visual change in color, that permits an uncomplicated explanation of results [23]. Also, the adaptable surface chemistry of AuNPs offers a remarkable linkability to a broad range of molecular probes with thiol groups for the functional combination and recognition of chemical and biological targets [24]. Choleva et al., described a paper-based apparatus in the form of a sensor patch (0.8 cm) which was able to determine antioxidant capacity through analyte-driven formation of AuNPs [25]. A seedbed of gold ions on a paper configuration of nanoparticles was used upon reduction by the presence of antioxidant complexes in liquid samples. The chromatic transitions from white or pale yellow to red, moving on the paper surface, were utilized to estimate the antioxidant strength of the solution. When the proposed method was examined against different antioxidant molecules, the strength of the color response was distinguished from high level to low level for catechin, gallic acid, caffeic acid, ascorbic acid, coumaric acid, vanillic acid, ferulic acid and cinnamic acid, respectively. The repeatability ranged between 3.58 and 6.62% and the reproducibility between 6.93 and 12.59%, suggesting the good accuracy of the assay. The linear range and LOD were 10 µM–1.0 mM and <1.0 µM, respectively. The Au-to-AuNPs assay illustrated several advantages compared to conventional antioxidant assays including portability, ease of use, simplicity, quick response, low price, high sensitivity, robustness and reproducibility, good stability, no requirement for specialized equipment, and no need for sample pretreatment and instrumental detectors [25]. In another study, AuNPs were synthesized through a mild chemical route which was based on water. The AuNP formation was achieved using polyphenols in olive oil and the reaction was followed by a sigmoidal curve [26]. The data illustrated that mixtures with *ortho*-diphenol functionalities actively reduced Au (III) to Au (0). A considerable relation among traditional approaches used to distinguish antioxidant activity (ABTS, DPPH and FC). The disadvantage of this method was that, in fat-rich samples, the developed method required the extraction of polyphenols by classical methods [26]. This disadvantage was solved by synthesizing the AuNPs using dimethyl sulfoxide (DMSO) as an organic solvent without the need for an extraction method. DMSO solubilized the sample and stabilized the AuNP suspension. It was able to act as a cryogenic preserver that prevent solidification at the temperatures used to quench the synthesis [27]. Tułodziecka et al., offered a method to evaluate the antioxidant capacity of *Brassica* (rapeseed) oilseeds, white flakes and meal extracts [28]. The analysis was based on the formation of AuNPs in an acetate buffer medium (pH 4.6). The antioxidant capacity of samples determined by the AuNP assay was compared with the FRAP, DPPH and FC methods. Remarkable effective correlations (R = 0.840–0.970) were found. High sensitivity and good repeatability were achieved. The linear range and LOD were 0.01–0.40 mM and 0.020 mM, respectively [28]. Other researchers designed a green synthesis of AuNPs in tea solution and fruit juices with auric tetrachloride (HAuCl_4_) without the addition of any other chemicals. Natural antioxidants were able to reduce Au^3+^ ions to form AuNP spheres with an average diameter of 22.9 nm and 12.8 nm in tea solution and orange juice, respectively. The advantage of this method was the properly regulated particle size, particle shape and an extremely limited size distribution [29]. An extraction-free method was described for the colorimetric determination of thymol and carvacrol [30]. The diagnosis of antioxidant activities was based on the formation of citrate-capped AuNPs and considered at two pH values, the first route was at pH 12, which agreed with the Folin-Ciocalteu method and the second route was between pH 12 and pH 9 for antioxidant isomer quantification. An advantage of this method was the ability to evaluate the quality of essential oils (fat rich samples) and is offers a valuable alternative to complicated, laborious and high time-consuming methods [30]. Scroccarello et al., reported an AuNPs-based polyphenol antioxidant capacity assay for the assessment of apple extract compositions. It was based on the ability of the polyphenol analytes to decrease Au^3+^. The AuNP formation depended on the analyte structure and concentration, which resulted in a red colored AuNP suspension. This assay required a preliminary mixing step, followed by metal nanoparticle formation under mild conditions. The linear range was 1–25 µM and LOD was ≤3.3 µM [31]. In another work, a heparin-stabilized AuNPs-based ‘cupric reducing antioxidant capacity’ (CUPRAC) colorimetric sensor was developed for tea antioxidant evaluation [32]. Heparin, as a sulfated polysaccharide, was the reducing agent as well as the stabilizing agent for defined negatively-charged AuNP synthesis. The resulting stabilized AuNPs were added to a copper(I)-neocuproine (Cu(I)-Nc) solution formed by the reaction of Cu(II)-Nc with antioxidants. The linear range was 3.1–90.5 µM and LOD was 0.2 µM. This developed sensor caused the enhancement of physico-chemical properties such as resistance to accumulation/aggregation [32]. In another work, researchers synthesized AuNPs and AgNPs using different reducing and capping agents. The functionality was based on the interactivity of the antioxidants with the nanoparticles which caused accumulation or morphological alterations leading to a change in the sensors’ colors. This method determined individual antioxidants as well as antioxidants in the mixtures incorporated with pattern identification and multivariate calibration approaches. Although the method was irreversible, it was a valid device for analysis of several antioxidants in real samples [33]. Seed-mediated growth is a suitable way in the managed synthesis of nanomaterials. Thus, several researchers have used Au nanorods (AuNRs) as seeds, which applied the seed-mediated growth of bimetallic NRs in antioxidant evaluation. Particularly, advancement of the analytical probe from AuNRs to Au@Ag core-shell NRs was well adjusted by seed-mediated growth, where little amounts of antioxidants played vital roles in this process. Actually, AuNRs were used as seeds for the epitaxial growth of the Ag, which could be generated from the redox reaction between antioxidants and silver nitrate (Figure 1a). The linear range was 0.01–30 μM and LOD was 0.0064 µM [34].

#### 2.1.2. AgNPs-Based Colorimetric Assays

AgNPs are other popular nanoparticles in colorimetric assays with some advantages such as cheapness, easy preparation, flexibility and very high extinction coefficients. The plasmonic band of AgNPs can be changed according to the size and distance, which makes them ideal for naked eye-distinguishable readout sensors [35]. They have several advantages compared to AuNPs. The extinction coefficients of AgNPs are higher than those of AuNPs for an identical average size, but AuNPs are more favored. Actually, AgNPs functionalization regularly leads to their chemical degradation and subsequently the AgNPs surface can be conveniently oxidized, thus decreasing their stability [36].

Recently, researchers have suggested a method based on poly (vinyl alcohol)-fixed AgNPs (PVA-AgNPs) and seed-mediated growth. In the presence of polyphenols, Ag^+^ was reduced to Ag^0^ and assembled on the PVA-AgNPs’ surface, leading to a growth in the particle size [37]. The increase in absorbance of the PVA-AgNPs suspension (with a negligible red shift) was correlated to the antioxidant capacity of ginger samples. The formation of AgNPs caused by gallic acid in a presence of PVA-AgNPs seeds is shown in Figure 1b. The linear range and LOD were 25–200 µM and 22.1 µM, respectively. The method was compared with the ABTS assay. Although the ABTS evaluation gave higher antioxidant values than the PVA-AgNPs approach, the values were correspondingly well ranked. This method was green due to the use of less toxic chemicals for synthesis of the particles and also the use of a biodegradable PVA polymer that was not detrimental to the environment [37]. In another study, a AgNPs-based spectrophotometric approach for antioxidant capacity evaluation was developed. The assay was based on the capability of tea polyphenols to decrease Ag(I) levels and stabilize the produced AgNPs(0) at 25 °C. This method showed good reproducibility (RSD ≤ 13) and was uncomplicated, sensitive and cost efficient [5]. In another study, Selvan and colleagues considered phyto-synthesis of AgNPs, by utilizing aqueous garlic, green tea and turmeric extracts [38]. Phytochemical assays showed the existence of high quantity of biochemicals in these extracts, which serve as reducing and capping agents for changing silver nitrate into AgNPs. The antioxidant activity was assessed by classical methods. The AgNPs synthesized by the green approach illustrated exceptional activity regarding the standard antioxidants ascorbic acid and rutin [38].

#### 2.1.3. CeONPs-Based Colorimetric Assays

Cerium nanoparticles (CeONPs, nanoceria) display numerous properties such as catalytic activity, fluorescence quenching, high surface area and oxygen movement ability [39]. Nanoceria is a fascinating material that can act as an oxidant as well as an antioxidant, because according to its preparation method and environmental circumstances, it can vary between trivalent and tetravalent oxidation states of cerium [40]. The uncommon redox and catalytic properties of CeONPs differ with the size, shape, charge, surface layer and chemical reactivity [41]. Nanoceria-based assays show several advantages such as easy operation, rapid detection, biocompatibility and low cost. On the other hand, they have some drawbacks such as low sensitivity and low stability for enzymatic-based approaches, however, they are highly stable at 25 °C in enzyme-free methods [39].

The antioxidant capacity of rapeseed was studied by using CeONPs [42]. The rapeseed antioxidants decreased cerium (IV) ions giving red-purple solutions of CeONPs. The method was usefully utilized at pH 5.6 (acetate buffer) and the resulting CeONPs appeared elliptical and rod-shaped. The antioxidant capacity of the extracts from real samples fluctuated between 1037 and 3012 μmol sinapic acid 100 g^−1^ and 3859–12,534 μmol sinapic acid 100 g^−1^ for CeONP and AgNP assays, respectively. Linear range and LOD were 1.2–1.7 mM and 2.75 µM, respectively [42]. In another work, a novel colorimetric sensor for an antioxidant activity assay was suggested by using poly (acrylic acid) sodium salt (PAANa)-coated CeONPs. PAANa-coated CeONPs oxidized 3,3′,5,5′-tetramethylbenzidine (TMB) in a slightly acidic solution to form a blue charge-transfer complex. PAANa was applied to attain long time utilization and avoid accumulation for the stabilization of nanoparticles (Figure 1c) [17]. Potential interferents such as citric acid, mannitol, glucose, sorbitol and benzoic acid did not negatively influence the antioxidant activity diagnosis. This sensor was low-cost, robust, highly sensitive and more selective than similar colorimetric sensors depending on the inherent color alteration of nanoceria. In another work, researchers reported a lightweight distance-based recognition paper device for rapid diagnosis of tea antioxidant activity by utilizing nanoceria. The analysis was based on limited reduction of cerium ion from Ce^4+^ to Ce^3+^. The lowest LOD was observed for epigallocatechin (4.0 μM) < gallic acid (5.0 μM) < caffeic acid, quercetin (6.0 μM) < ascorbic acid, and vanillic acid (8.0 μM). The sensor had high yield and tolerance limit and was stable for 50 days at room and low temperatures [11]. In a simple method, a well-organized microfluidic paper-based analytical device (μPAD) with imaging abilities with adapted for antioxidant analysis. A simple way for μPAD production through the application of clear nail paint led to formation of hydrophobic hurdles and well-defined channels. The μPADs-infused poly (methacrylic acid) (PMAA)-coated CeONPs oxidized TMB, leading to the formation of a blue-colored charge-shift complex. This sensor could be stored for a long time without losing the activity [43].

#### 2.1.4. Other Nanomaterial-Based Colorimetric Assays

In addition to widely used AuNPs and AgNPs, some other nanomaterials show the potential for application in the analysis of antioxidants. Iron oxide nanoparticles (IONPs) with several advantages including chemical stability, nontoxicity, low cost and ease of utilization have been used for determining the antioxidant capacity evaluation of foods [44].

Szydłowska-Czerniak and colleagues evaluated the antioxidant capacity of rapeseed oils using IONPs. Ferric ions were decreased by oil extracts in acidic medium by formation of yellow solutions of IONPs [44]. The IONPs had a sphere shape and were homogeneous. This method did not need specialized equipment and particular reagents. However, the offered approach seemed to be less sensitive than the modified ferric reducing antioxidant power (FRAP) and 2,20 diphenyl-1-picrylhydrazyl (DPPH) methods. It showed an acceptable intra-day precision in comparison to the modified FRAP and DPPH approaches. Thus, the novel method could be a substitute for the modified antioxidant capacity assays [44]. A sensitive solid membrane optical sensor was proposed for evaluating the antioxidant capacity of fruit juices [10]. The functionality of the sensor was dependent on immobilizing a chromogenic oxidant, Fe(III)-o-phenanthroline (Fe(III)-phen), onto a Nafion cation-exchange membrane. The colorimetric evaluation was done through the reaction of antioxidants with formation of highly-colored Fe(II)-phen. The linear range and LOD were 2.45–47.39 µM and 0.26 µM, respectively. The sensor was more sensitive than the solution-phase method because the membrane concentrated the color from a larger volume solution. Also, this sensor was easily adaptable to a kit format [10].

Wu et al. used three types of nanoparticles, including AuNPs, AgNPs and IONPs, to assess the antioxidant capacity of Chinese rice wine and zhuyeqing liquor. The colorimetric methods showed good correlations with FRAP assays (the correlation coefficients were 0.952, 0.948, and 0.969 for AuNPs, AgNPs and IONPs, respectively). This method did not need expensive radical compounds and organic solvents [45].

MnO_2_ nanoparticles also have been regarded the most suitable inorganic materials due to their abundance, fine catalytic activity, and low price. MnO_2_ nanostructures have been extensively utilized in different sectors such as catalysis, electrochemical studies, and biological applications [46].

A detection method was developed for red wine antioxidants using the oxidase-like activity of two-dimensional MnO_2_ nanosheets [47]. Especially, the MnO_2_ nanosheets could change the colorless substrate TMB to a deep blue product (oxTMB) via catalytic oxidation. The linear range and LOD of MnO_2_ nanosheets were 3–15 µM and 0.3 µM, respectively [47].

In another study, researchers suggested a visual colorimetric sensor for the identification of antioxidants in serum based on a MnO_2_ nanosheets-TMB multicolor chromogenic system. The reaction between TMB and MnO_2_ nanosheets was blocked by the antioxidants because of the presence of the competitive reaction of MnO_2_ nanosheets and antioxidants [48]. In another work, gum arabic was used as the reducing and templating agent for the synthesis of MnO_2_ nanosheets [46].

MnO_2_ nanosheets were used as anoxidizing agents to oxidize TMB to a blue product. Because of the redox reaction between MnO_2_ nanosheets and antioxidants, the reduction in color intensity of the MnO_2_ mixture was observed in the presence of antioxidants. The LOD was 0.1 µM [46]. A nano-manganese oxide (nano-MnOx)-based spectrophotometric approach was proposed for estimation of antioxidant compounds. Actually, in the presence of hydrophilic and lipophilic antioxidants, the color intensity of TMB cations (TMB^+^) was decreased. This method was economical, easy to use, rapid, highly sensitive and with good precision [49].

Other nanomaterials which have been used for antioxidant determination in foods include Pt, Cu, rhodium and lanthanides. Romero et al., evaluated the antioxidant capacity of tea extracts using the DPPH, CUPRAC and two electrochemical approaches involving radicals generated from hydrogen peroxide on Hg and glassy carbon electrodes protected with PtNPs and polyneutral red (PNR-Pt). The LOD of this method was 17.2 µg·g^−1^ [50]. In another work, a new and rapid way for estimating total phenolic mixtures in tea and fruits was described utilizing colorimetric spots and a digital image-based (DIB) approach. The formation of colorimetric spots was done by reaction of diazotized aminobenzenes (sulfanilic acid, sulfanilamide, and aniline) with phenolic compounds in the extract to form an azo dye. This method was quick, cheap, flexible, robust and lightweight [19]. In a simple assay, the use of a paper-based device as a detection platform for determination of antioxidant activity in tea, wine and fruit juice was reported. Two antioxidant activity assays including ABTS and CUPRAC and one total phenolic content analysis including Folin Ciocalteau reagent (FC) assay were simultaneously used. The device consisted of a central sample zone joined to four pretreatment zones and successive detection zones to locate all three analyses and a sample blank measurement (Figure 2a,b). The linear range was 3–13 mM [15]. In another work, a colorimetric sensor including three lanthanide ions (Eu^3+^, La^3+^, and Sm^3+^) as sensor elements and Eriochrome Black T (EBT) as signal readout was developed. EBT and lanthanide ions formed binary networks, which led to a change of color from blue to pink. By incorporation of antioxidants, the sensor array showed cross-reactive replies to antioxidants (Figure 2c). This sensor was facile, robust and changes in color and absorbance were observed from the competitive binding from EBT and antioxidants to lanthanide ions [51]. A small, cheap and portable sensor for evaluating the whole antioxidant capacity in tea using cupric-neocuproine (Cu(II)−Nc) immobilized into a polymethacrylate matrix (PMM) with spectrophotometric quantification was developed. The absorption alteration related with the formation of the colored Cu(I)-Nc chelate in the PMM as a consequence of reaction with antioxidants [52]. The formation of colored spots by polyphenolic compounds through reaction with nano-oxides of Al_2_O_3_, ZnO, MgO, CeO_2_, TiO_2_ and MoO_3_ infused on filter paper was used for antioxidant analysis [53]. In another work, polyphenolic mixtures of tea were determined based on their reactions with citrate-capped rhodium nanoparticles (RhNPs). The linear range and LOD were 50–500 µM and 29 µM, respectively. This sensor showed high stability and good reproducibility. The results were correlated with the frequently used approaches (i.e., Folin-Ciocalteu and aluminum interaction analysis) [54]. Aid and colleagues reported a colorimetric paper microzone assay for analysing the total phenolic content of lime fruit in five imidazolium-based ionic liquid solutions with concentrations ranging from 50–100 mM. The formation of deposits of mixtures of the polyoxometalate-imidazole ionic liquid stopped the spectrophotometric diagnosis of analytes. The linear range and LOD were 0.25–2 mM and 0.08 mM, respectively [55]. For identification of whole polyphenol content of coffee samples, an interesting approach was proposed. A polyphenol sensor dependent on co-immobilization of NaIO_4_ and MBTH in paper as a test strip was devised. The sensor demonstrated sensitive responses to chlorogenic acid by forming a pink color. The linear range and LOD were 0.07–0.71 mM and 0.002 mM, respectively. The advantages of this sensor were good reproducibility, good stability (27 days at 4 °C), fast response, easy to operate, low-cost and reliability [56]. In another work, a colorimetric sensor array based on the reactions between TMB and metal ions (Ag^+^, Au^3+^, and Cr^6+^) as sensing receptors and the interactions between antioxidants and oxidized TMB (oxTMB) was reported. The oxidized form of TMB (blue oxTMB) produced by metal ions that could oxidize colorless TMB [6]. Representative examples of recent developed colorimetric assays for the determination of antioxidants have been listed in Table 1.

#### 2.1.5. Advantages, Limitations, and Potential for Practical Applications

The development of new sensors with a focus on the food sector is one of the vital areas for nanobiotechnology and nanomaterial science. Particular focus has been given to methods with high sensitivity, quickness, low requirement of sample, required simple and low cost instrumentation, with a purpose to rationalize the use of research resources [18]. In this regard, nanomaterials have been employed in numerous polyphenol sensors and sensing schemes. Nanomaterials have special thermal, mechanical, optical, electrical, magnetic and biological properties, which are size-dependent and can be tuned by simply adjusting the size, the shape and the extent of agglomeration. Nanomaterials are utilized as catalytic instruments, immobilization platforms or as optical or electroactive labels to enhance the sensing performance revealing higher sensitivity, stability, and selectivity [57]. Nanoparticle-based colorimetric sensors have particular advantages such as tiny size and great particular surface area, good reactivity, implanting process ability into narrow films, paper and other matrices that are able to combined to optical apparatus appropriate for automated assay development [12]. Moreover, they have several advantages compared to conventional antioxidant assays including portability, ease of use and uncomplicated operation, fast responses, low price, high sensitivity, robustness, reproducibility, long-term stability, no requirement for specialized equipment, and minimal need for sample pretreatment [25]. The main disadvantage of colorimetric sensors is that, the sensitivity of colorimetric approaches is lower than that of other methods such as fluorescence. Thus, signal amplification should be considered to enhance the sensitivity for identification of low concentrations materials [58].

A comparison of different nanomaterials by researchers showed that the CeONPs-based methods illustrated an acceptable precision (RSD = 1.2–3.9%) compared with that of the AgNPs-based methods (RSD = 0.5–4.2%), also, higher sensitivity of sinapic acid (ε = 1.24 × 10^4^ Lmol^−1^ cm^−1^) for CeONP and (4.1 × 10^3^ Lmol^−1^ cm^−1^) for AgNP. Therefore, the CeONPs-based approaches were a replacement strategy for these, depending on the formation of metal nanoparticles such as AgNPs, and they could be used by oil industry laboratories for the antioxidant capacity evaluation of oilseeds, semi-products, end products, and by-products [42]. Also, another application in the oil industry was reported by using iron oxide nanoparticles for colorimetric assays. It was an uncomplicated and low cost method, which did not need exclusive equipment for evaluating the antioxidant activity of oils and the modification of the refining procedure [44]. Furthermore, Gatselou et al., compared the antioxidant capacity of different nanoparticles [54]. Results showed that RhNPs-based methods enabled one to do analyses without inflexible timing restrictions in comparison to other nanoparticle-based analysis such as AgNPs, which required extended incubation times [59] and also such as AuNPs, that required severe control of the reaction time [60] or usage of high temperatures to stimulate redox kinetics in order to reach an equilibrium [61]. In another study, a comparison of sensors with solution-based methods was reported [10]. They reported that the solid membrane optical sensor was more sensitive than the solution-phase ferric-phenanthroline approach because the membrane concentrated the colored strains from a greater volume of solution [10]. In addition, in another study, the solid-state approach suggested by researchers had some advantages over solution-based assays such as no need for sample pretreatment. Also, usage of PMM-Cu(II)-Nc in a solid state method was appropriate for colored or opaque samples which were not measurable with solution-based approaches [52].

### 2.2. Fluorescence Assays

Fluorescence-based assays use emission intensity, wavelength, fluorescence lifespan, or fluorescence anisotropy as analytical data. Many factors interfere in driving of signals that are including, changes in pH, charge, polarity, or viscosity of fluorophores. Fluorescent biosensors utilize organic dyes, carbon and graphene quantum dots (CDs and GrQDs, respectively), and semiconductor QDs as fluorophores [62]. They have attracted more focus due to their clarity, convenience, high sensitivity, high-output, fast response, simplicity of automation, and minimized background signals [63,64]. Their limitation is the requirement of specific instrumentation for reporting their results, which is not economical.

#### 2.2.1. Quantum Dots (QDs)-Based Fluorescence Assays

QDs are emerging nanomaterials with many applications in analytical chemistry. QDs are semiconductor nanocrystals with size-dependent fluorescence properties [18]. In contrast to conventional organic fluorescence probes, QDs have many superior features with respect to wideband excitation, narrow bandwidth and high intensity emission [65].

In biological applications, QDs as well as fluorophores have become important [66], because of the possibility to size-tune fluorescent emissions as a consequence of the core size, shape and material [22]. Dwiecki et al., suggested an approach for total phenolic compounds identification of common drinks (tea and coffee) based on CdTe QDs fluorescence in the presence of an analyte [67]. Polyphenols acted as reducing agents with the ability to transfer electrons to the CdTe-sodium periodate system leading to perturbation of the conduction of excited electrons from QD to the acceptor molecules happening in the absence of polyphenols. This method showed higher sensitivity in comparison with the Folin-Ciocalteu approach and lower impact of intervention (derived from proteins and reducing sugars) on the outcomes. The linear range and LOD were 0–4.24 µM and 0.63 nM, respectively [67]. In another study, the antioxidant evaluation dependent on the redox alteration of polyaniline (PANI, a conducting polymer) was reported. Actually, the emeraldine base (EB) of PANI fibers could be reduced to the leuco-emeraldine base (LB) form in the presence of antioxidants, inducing a color change (from purple to light gray). Furthermore, the EB configuration of PANI could accurately quench the fluorescence of CdTe quantum dots, and the fluorescence was recovered with incorporation of an antioxidant. The linear range and LOD were 2–40 µM and 100 nM, respectively. This method was simple, sensitive and label-free [68]. In another study, an approach for the diagnosis of the antioxidant activity of juice beverages based on photocreation with visible radiation of radical species from CdTe QDs capped with L-glutathione and utilizing luminol as a chemiluminescence probe was proposed. The linear range and LOD were 0.01–5 µM and 0.5 µM, respectively [69]. In another work, a 3-dimensional sensing chip, constituted of CdSe/ZnS QDs and graphene, was suggested to determine antioxidants depending on the concurrent use of the fluorescence, electrochemical and mass-sensitivity characteristics of the nanocomposites. The focus was on the numeral of phenolic hydroxyl groups on the analytes so as to be adsorbed on the sensing nanochip, leading to fluorescence quenching of the QDs [70].

Luminescent blue GrQDs were used as sensing probes in a paper-based sensing apparatus with smartphone readout for antioxidant estimation in wine. Different levels of GrQDs quenching were achieved in order of morin > myricetin > quercetin > kaempherol. The 3D-printed apparatus with a dark chamber contained a strip hole where the paper strip went through. Each spot was processed, one at the time, to reach the UV LED area. Phenolic mixtures susceptible of producing GQDs quenching and those couldn’t produce it were denoted by a yellow and red circle, respectively (Figure 3a). This approach was uncomplicated, inexpensive and quick [71]. In another study, an uncomplicated, sensitive and label-free GrQDs-based fluorescence sensing system was used for diagnosis of ascorbic acid in the presence of copper ions [72]. Because of the well-organized electron-transfer between GrQDs and Cu^2+^ ions, the fluorescence of GrQDs was considerably quenched by Cu^2+^ ions (Figure 3b). The linear range and LOD were 0.3–10 µM and 0.094 µM, respectively [72]. In another study, a fluorescence sensor for the detection of ascorbic acid (AA) in fruit juices was suggested based on the fluorescence resonance energy transfer (FRET) between GrQDs and squaric acid (SQA)-iron(III). In this analysis, iron(III) could quickly react with the SQA to form SQA-iron(III). By oxidation-depletion between iron(III) and AA, the fluorescence of GQDs could be sensitively turned on by AA. The linear range and LOD were 1–95 µM and 0.2 µM, respectively [73]. Also, by using of the orange emission GrQDs, a fluorescence turn-on analysis for identification of ascorbic acid (AA) was reported. Catechol could be oxidized by hydroxyl radicals that were produced by horseradish peroxidase (HRP) and H_2_O_2_ and this process led to the conversion of catechol to *o*-benzoquinone, which could considerably quench the fluorescence of GrQDs (Figure 3c). However, in the presence of AA in the system, it could consume a portion of the H_2_O_2_ and hydroxyl radicals, inhibiting the creation of *o*-benzoquinone, and consequently, in fluorescence recovery. The linear range and LOD were reported to 1.11–300 µM and 0.32 µM, respectively [74]. A GrQDs-hypochlorite system was used to determine non-enzymatic and enzymatic antioxidants in commercial drinks. The recognition basis depended on the fact that antioxidants could preserve the fluorescence of GrQDs from hypochlorite-caused quenching by acting as hypochlorite scavengers. This system illustrated an outstanding analytical outputs for commercial drinks (>89.9%) and good comparability with ELISA testing for superoxide dismutase secretion in a cell-conditioned medium. The linear range and LOD were 8–60 µM and 1.4 µM, respectively [75].

GrQDs were used as practical fluorescent probes for the identification of chromium(VI) and ascorbic acid in an on-off-on mode. The reason for the strong quenching of GrQDs fluorescence by Cr(VI) was due to an internal filter impact and static quenching. The fluorescence of GrQDs-Cr(VI) system was changed back to Bon by adding ascorbic acid which reduced the yellow Cr(VI) ion, resulting in removal of the internal filter impact and static quenching. The linear range and LOD were 0.05–500 µM and 0.0037 µM, respectively [76]. In another study, a switch-on fluorescence sensor for glutathione (GSH) determination in food samples was outlined. A graphitic carbon nitride quantum dots (g-CN QD)-Hg^2+^ chemosensor was utilized in this technique. The fluorescence signal was quenched by Hg^2+^. GSH and Hg^2+^ showed a competitive tendency to react with the functional groups on the surface of g-CN QDs, resulting in switching of the fluorescence sensor to the “on” state. The linear range and LOD were 0.16–16 µM and 37 nM, respectively. The advantages of this approach were high reactivity and sensitivity, low price and speed [77]. Also, a sensitive and selective spectrofluorometric approach for diagnosis of flavonoids indicated as ‘quercetin equivalents’ in apple juices was reported by other researchers. The linear range and LOD were 1.5–60.5 mg L^−1^ and 0.3 mg L^−1^, respectively [78].

Representative examples of recent developed fluorescence assays for the determination of antioxidants are listed in Table 2.

#### 2.2.2. Advantages, Limitations, and the Potential for Practical Applications

As can be seen, QDs have been extensively utilized in fluorescence-based analysis for antioxidants analysis. Compared to conventional organic fluorescence probes, QDs have much superiority with respect to wideband excitation, limited bandwidth and high power emission. QDs have become used mainly as fluorophores in biological applications, because of the probability to size-tune fluorescent emission as a basis of the core size, shape and material [18,66]. Although fluorescence methods are sensitive, they require detection by fluorimeters, which may not be routinely available in analytical laboratories. However, this tool is generally utilized in many cell culture laboratories. Furthermore, the long analysis time (around 1 h) has also been a crucial criticism, but this restriction has been partly controlled by evolution of high-throughput analysis [79]. Unlike inorganic QDs, GrQDs have attracted enormous attention in biosensing, because of their higher stability, photoluminescence quantum output, lower cytotoxicity and good biocompatibility [80]. GrQDs have been suggested for the diagnosis of heavy metals, small molecules, and biomacromolecules [81].

In a study conducted by Rodrigues et al., CdTe QDs capped with L-glutathione and luminol were used as a chemiluminescence probe. They reported the requirement of very low consumption of reagents in this method compared to the photo-bleaching method [69]. In another study, a FRET-based nanosensor with fluorescence turn-on analysis for the ascorbic acid recognition was reported [73]. Compared with a similar study [76] that used GrQDs/CQDs for detecting ascorbic acid, the FRET-based nanosensor method did not need the usage of highly toxic metals (e.g., Cr(VI)) and overpriced enzymes.

The fluorescence “turn off–on” mode of this sensor had the benefits of adjustability and high selectivity. Furthermore, this FRET-based sensor did not need any surface modification of GrQDs or organizing of any covalent join between the acceptor and the fluorophore, or providing substantial flexibility and simply in the probe manufacture.

## 3. Electrochemical Sensors and Biosensors

Electrochemical sensing strategies have attracted great attention in the determination of different analytes due to the rapid, sensitive, accurate and low-cost analysis they provide. Regarding these properties, electrochemical sensors can be considered as ideal analytical tools for the direct analysis of antioxidants and measurement of total antioxidant capacity (TAC) of foodstuffs. Electrochemical sensors and biosensors for antioxidants’ analysis have been designed using different types of electrode, transducers and receptors. In some cases, nanomaterials have been integrated into biosensors to obtain improved performance and higher sensitivity. In this section, electrochemical sensors for antioxidant monitoring are categorized and discussed based on different kinds of receptors including enzyme, cell, DNA, and molecularly imprinted polymers (MIPs). Moreover, nanozyme-based electrochemical sensors for antioxidant analysis are introduced and discussed.

### 3.1. Enzyme-Based Electrochemical Biosensors

Enzyme-based electrochemical biosensors use an enzyme as the bioreceptor for the identification of a target molecule based on the principle of inhibition of enzyme activity. By enzyme exposure to a specific substrate with inhibition activity at a given time, the progression of the enzymatic reaction is inhibited and the target analyte is quantified by deciding the relationship between the enzyme hindrance rate and inhibitor concentration [2]. Enzyme-based biosensors possess several advantages related to the enzyme nature. They are highly selective for a specific substrate, and a large number of substrate molecule reactions can be catalyzed only with a single enzyme molecule which results in an amplification effect and increased sensitivity [82]. Conversely, electrochemical biosensors based on enzyme-catalyzed reactions are simple and widely available. The most common enzymes used in biosensing belong to the oxidoreductase, hydrolase and lyase groups. In the case of antioxidant analysis, proteases such as tyrosinase [83], peroxidase [84] and laccase [85] have been utilized in the evolution of electrochemical biosensors. In this regard, antioxidant activity can be measured via biochemical oxidation subsequently by electrochemical reduction. The electric connection of oxidoreductase and the electrochemical transducer shows good properties and the analysis is performed by managing the enzyme reaction in real-time [18]. Tyrosinase and laccase are the two most extensively utilized protease enzymes for antioxidants’ analysis, particularly phenolic compounds evaluation [86,87,88].

There are numerous studies on enzymatic electrochemical biosensors for antioxidants analysis. Most of them have been designed based on a complex platform by integrating a variety of nanomaterials. The type of enzyme used in these biosensors is based on the analyte specificity, while the nanomaterial can enhance the electrical conductivity and performance of biosensor. In addition to nanomaterials, other biomaterials such as polymer membranes and gels are used to increase the biosensor performance.

#### 3.1.1. Peroxidase-Based Electrochemical Biosensors

Peroxidases are enzymes that catalyze oxidation-reduction reactions by free radical mechanism. They transform substrates into oxidized or polymerized products. HRP is one the most commonly used peroxidases in biosensing applications and biochemistry. There are only a few studies on peroxidase-based electrochemical biosensors for antioxidants analysis. Wu et al. immobilized HRP on Au-Pt nanotube/Au graphene for concurrent electrochemical diagnosis of BHA and propyl gallate (PG) [84]. In this study, a carbon electrode was first modified with gold nanoparticles-graphene (AuNPs-Gr) hybrids. Then, HRP was immobilized onto the modified electrode via electrostatic attraction (Figure 4a). Eventually, the spiny Au-Pt nanotubes were trimmed on AuNPs-Gr hybrids to generate a matrix nanostructure. This web-like nanostructure both speeded up the electron shift and trapped the HRP enzyme. Under the optimal conditions, BHA and PG showed distinctive oxidation waves by linear-sweep voltammetry (LSV) test. The proposed biosensor showed LODs of 0.046 and 0.024 mg L^−1^ for BHA and PG, respectively. As a result of combining AuNPs-Gr hybrids with unique physical and electrical properties and high effective surface area, and Au-Pt bimetallic nanotubes with excellent catalytic properties, the fabricated sensor exhibited increased sensitivity, stability and reproducibility. Peanut oil, potato chips and cookies were used as food matrices for the analysis of BHA and PG by the biosensor.

#### 3.1.2. Laccase-Based Electrochemical Biosensors

Laccases are multicopper oxidoreductase enzymes with the capability to oxidize a numerous of phenolic compounds like polyphenols, *ortho*- and *para*-diphenol groups, aminophenols and methoxyphenols. They can also oxidize polyamines, aromatic amines and lignins. Laccases can be isolated and purified from bacteria, fungi and plants [88]. Laccase exhibits a good stability among redox enzymes which make it ideal for antioxidants analysis. de Oliveira Neto et al. developed a laccase-based modified carbon paste biosensor for the determination of total phenolic content (TPC) and antioxidant capacity (AOC) of honey [85]. Electrochemical variables including peak current and peak potential were achieved by differential pulse voltammetry (DPV). The results obtained by the biosensor exhibited acceptable association with the spectrophotometric FRAP and DPPH radical scavenging assays. The assay was rapid with detection time of <30 s, in accordance with the time for enzymatic oxidation of phenolic mixtures.

Like peroxidase-based biosensors, nanomaterials can be integrated into laccase-based biosensors to improve sensitivity. Zrinski et al. immobilized laccase onto AuNPs/graphene nanoplatelets-modified screen-printed carbon electrode (AuNPs/GNPI-SPCE) [89]. The modified electrode was utilized for amperometric diagnosis of hydroquinone (HQ) and other phenolic compounds. GNPI (a 2D carbon nanomaterial) with better thermal, mechanical and electrical features than other carbon nanostructures, act as “electronic wires”. These wires provide shorter shift of electrons of prosthetic groups located in the structure of the enzyme deeply and secure the protein from adsorptive denaturation on electrodes or undesirable inclinations of molecules. This characteristic makes them ideal substrate for the immobilization of redox enzymes and fabrication of electrochemical biosensors. AuNPs/GNPI accelerated the electron shift between the electroactive site of enzyme and electrode surface and facilitated the orientation of the molecules to determine phenolic mixtures. The proposed biosensor illustrated a wide linear range for HQ from 4 to 130 µM with a LOD of 1.5 µM. The biosensor, with good repeatability, reproducibility, long-lasting stability and high selectivity towards HQ, was used for the diagnosis of AOC in wine and blueberry syrup. The results were comparable with those from the conventional spectrophotometric Trolox equivalent antioxidant capacity (TEAC) assay.

In order to increase the available area for laccase immobilization, AuNPs were electrodeposited onto SPCE modified with polypyrrole by in-situ electropolymerization (Figure 4b) [90]. In the presence of propolis extract containing polyphenolic compounds, immobilized laccase oxidized polyphenols; subsequently, these compounds were reduced on the surface of modified electrode by amperometry at −450 mV. A linear response was obtained in the concentration range from 1 to 250 μM expressed as caffeic acid, with a LOD of 0.83 µM. The analysis time was only 15 min which was much less than the time of Folin-Ciocalteu spectrophotometric method (85 min). The biosensor showed high selectivity, long-term stability (one month at 4 °C), good reproducibility, portability, low-cost, high accuracy and wide linear range for detecting polyphenols in propolis samples.

#### 3.1.3. Tyrosinase-Based Electrochemical Biosensors

Tyrosinase is a copper-containing oxidase which catalyzes two oxidation reactions employing oxygen: (1) *o*-hydroxylation of monophenolic compounds to *o*-diphenol compounds due to its monophenolase or cresolase activity, and (2) oxidation of *o*-diphenolic compounds to *o*-quinones through its diphenolase or catecholase activity. Tyrosinase can act on monophenols as well as diphenols as substrate [91]. Tyrosinase is present in plant, animal tissues, bacteria, fungi and insects. Tyrosinase has been immobilized on different kinds of electrodes as well as in combination with a variety of nanomaterials for developing electrochemical biosensors for antioxidant analysis. Different kinds of nanomaterials including metallic and metal oxide nanoparticles (e.g., gold, silver and platinum), carbon nanostructures (e.g., carbon nanotubes, graphene and carbon black) and semiconductor quantum dots have been employed to increase the performance of tyrosinase-based electrochemical biosensors. Graphene-based materials have great potential for developing biosensors. In order to enlargement the electrode surface along with the electron transfer rate, a composite of graphene oxide (GO) and multi-walled carbon nanotubes (MWCNTs) was utilized to modify glassy carbon electrode (GCE) and fabricate an amperometric biosensor for polyphenols detection [92]. Before enzyme (laccase or tyrosinase) immobilization on the surface of modified electrode, GO was reduced by an electrochemical method based on cyclic voltammetry as an environmentally friendly method. Reduced GO (rGO) exhibited greatly better electrical conductivity than GO. On the other hand, MWCNTs with interesting electrical properties acted as molecular wires of oxidase enzymes to provide more well organized amperometric biosensor. The enzyme immobilization on the surface of modified GCE was tested using three reagents including Nafion, chitosan, and, bovine serum albumin (BSA) cross-linked with glutaraldehyde in order to find the best system for the long-lasting stability of the enzyme and thus the longer stability of the biosensor during storage. The best condition to immobilize the enzyme was found to be BSA cross-linked with glutaraldehyde for laccase, and chitosan for tyrosinase. The laccase-based biosensor showed a higher operational stability (retaining 93.3% of its initial sensitivity after one month) compared to the tyrosinase-based biosensor (two days) which was related to the unstable nature of tyrosinase. The LODs towards catechol were evaluated to be 0.3 µM and 0.5 µM for laccase-based and tyrosinase-based biosensors, respectively. The biosensor was employed for the detection TPC in fruit juices.

GO with its high surface to volume ratio, good biocompatibility, good dispersion and amphiphilic nature (providing both water solubility and interaction with other compounds) is a potential matrix for electrochemical sensors. In a tyrosinase-based electrochemical biosensor, GO was used to modify GCE [93]. Then, tyrosinase was immobilized on the surface of modified GCE through glutaraldehyde (Figure 4c). The fabricated biosensor exhibited a low LOD of 0.03 µM and a wide linear range from 0.05 to 50 µM catechol concentration. Moreover, the biosensor dispalyed good reproducibility and repeatability, high selectivity and long-lasting stability (retaining 77% of its primary current response after one month). The total analysis time consisting sample preparation, sample measurement and information processing was less than 1 h.

The extensive majority of the studies on electrochemical biosensors using polyphenol oxidases, have employed amperometric transducers. However, electrochemical transducers dependent on potentiometric measurements exhibit several advantages including uncomplicated electronics and a high sensitivity. In this regard, a label-free potentiometric biosensor using tyrosinase was developed for diagnosis of total phenols in honey and propolis samples [94]. A solid-contact transducer was fabricated consisting two layers. The first layer contained a combination of poly(vinyl) chloride carboxylated (PVC-COOH), graphite and potassium permanganate. The second layer containing a blend of PVC-COOH and graphite was put down on the first layer. Tyrosinase was immobilized on the surface of developed solid-contact transducer via reaction with N-(3-dimethylaminopropyl)-N’-ethylcarbodiimide hydrochloride. The proposed biosensor exhibited a LOD of 0.73 µM with a broad linear range from 0.93 µM to 8.3 × 10^−2^ M towards catechol. The biosensor exhibited good similarity with the results achieved by the Folin-Ciocalteu method. However, the developed biosensor was faster and showed good selectivity, high mechanical resistance, long-term stability and re-usability during three months. Moreover, the biosensor can be easily miniaturized for on-site determination of phenolic compounds.

#### 3.1.4. Advantages, Limitations, and the Potential for Practical Applications of Enzyme-Based Biosensors

Enzyme-based electrochemical biosensors are among the most advanced and financially successful analytical tools due to high catalytic activity and selectivity of enzymes, along with financial accessibility of purified enzymes. The most significant progress in the field of enzyme-based biosensors is associated to the immobilization of the bioreceptor on the electrode surface. Enzyme-based electrochemical biosensors show good performance and high efficiency for practical applications. However, several parameters should be considered before commercialization of an enzyme-based electrochemical biosensor for measurement of phenolic combinations and antioxidant capacity analysis. In order to enzyme immobilization with high effectiveness and long-lasting stability, different kinds of nanomaterials and polymer membranes can be integrated into biosensors. They must be carefully selected based on the desired effects. Nanomaterials can be used in several ways in order to enhance analytical properties of the biosensor. They can be either co-immobilized with the enzyme or combined with the transducer [95]. Nanomaterials, with their high surface-to-volume ratios, expand the available area for more efficient enzyme immobilization. Furthermore, they are able to accelerate the electron transfer resulting in enhanced sensitivity of biosensor, lower LOD and lower detection time. Furthermore, surface of nanomaterials can be conveniently functionalized with a variety of chemical groups which is necessary for the interaction with biomaterials in biosensors [96]. Another important parameter is the utilization of biocompatible materials for enzyme immobilization on the electrode surface which leads to increased enzyme stability, biosensor stability during storage and increased sensitivity. Matrix interference is a main challenge in developing different kinds of biosensor which can affect the sensor’s sensitivity and stability. Innovative approaches in sample pretreatment and optimization of the sensor’s sensitivity can overcome this problem. The selection of an appropriate enzyme according to the type of analyte is very important to design an enzyme-based biosensor. Because of the high selectivity of enzymes, one type of enzyme cannot detect all antioxidant compounds. For instance, laccase cannot catalyze oxidation of phenolic compounds with amine group in the meta position such as 3-amino phenol or other monophenols.

### 3.2. Cell-Based Electrochemical Biosensors

Cell-based biosensors are analytical tools composed of living cells as bioreceptor. In electrochemical biosensors, any physiological change in the cell can be detected by an appropriated transducer which converts these changes into a quantifiable electrical signal [97]. Cell-based biosensors have been recently developed for sensing of toxic materials in food and environment, as well as for drug screening and medical diagnosis [98]. Different types of cells including mammalian cells, plant cells, microbial cells, and their recombinant kinds can be utilized as bioreceptors for developing electrochemical biosensor. Among them, mammalian cells are good candidates for antioxidant analysis due to production of responses at the cellular level. In addition, microbial cells including bacteria and yeasts can be a good option for this purpose because of numerous advantages such as rapid growth, low price, simple cultivation, simplicity of genetic manipulation, easy accessibility and capability to metabolize a variety of substrates [99]. Cell-based biosensors for antioxidant analysis provide results based on the impacts of antioxidants on the whole cell.

Reactive oxygen species (ROS) are a group of the most important cellular compounds which can be monitored in cell-based electrochemical biosensors for antioxidant analysis. ROS are highly reactive molecules produced in vivo during metabolic processees. The agglomeration of surplus amounts of ROS in cells, as a consequence of environmental stress, leads to injury to cell components such as nucleic acids, proteins and lipids. In addition to cellular antioxidative systems, exogenous synthetic and natural antioxidants are able to remove free radicals and other pro-oxidants. Macrophage cells with natural ability to detect oxidative stress are good candidates as bioreceptor in cell-based electrochemical biosensors. For instance, RAW264.7 macrophage cells were used in a cell-based electrochemical biosensor for determining the antioxidant capacity of cell-free extracts from *Lactobacillus plantarum* strains separated from Chinese dry-cured ham (Figure 5a) [100]. The detection was based on the production of ROS by RAW264.7 cells (after stimulating with phorbol 12-myristate 13-acetate) in the cytoplasm which caused the release of H_2_O_2_. The H_2_O_2_ released from immobilized RAW264.7 cells on the surface of electrode can be recognized quickly and transformed to a sensitive electrochemical signal. In order to receive ideal cell adhesion and to preserve cell viability, alginate/graphene oxide hydrogel was used for the encapsulation of RAW264.7 cells. The encapsulated cells were immobilized on the surface of acidified manganese dioxide (a-MnO_2_)-modified gold electrode. These nanoparticles enhanced the catalytic characteristics and decreased the charge-transfer resistance in order to receive a higher sensitivity. In this study, the released H_2_O_2_ was adsorbed by the a-MnO_2_-modified gold electrode and catalyzed at the active areas on the surface of MnO_2_. Therefore, MnO_2_ decreased and electrooxidized at the electrode surface. The oxidation current significantly raised with the prodution of H_2_O_2_. Extracts from *L. plantarum* strains at the population of 10^10^ CFU mL^−1^ exhibited the highest antioxidant capacities. The biosensor showed a LOD of 0.02 µM with a linear range of 0.05–0.85 µM. The same authors used the Caco-2 cells to fabricate an electrochemical biosensor for estimating antioxidant capacity of Asp-Leu-Glu-Glu separated from dry-cured Xuanwei ham [101]. In this study, Caco-2 cells confined in alginate/graphene oxide hydrogel were immobilized on the gold electrode modified with platinum NPs (PtNPs) and silver nanowires. PtNPs improved electrocatalytic current and increased electrode sensitivity for H_2_O_2_ detection. On the other hand, silver nanowires not only exhibited excellent catalytic activity for H_2_O_2_ reduction but also minimized inhibiting signals from other molecules such as ascorbic acid and uric acid. Under optimal condition, the LOD was 0.12 µM with a linear range of 0.2–2 µM. Differences in the type of cell used or the nanomaterials employed to modify the electrode could be the reasons for the increased LOD compared to the previous study.

In a similar study, Ye et al. used human lung adenocarcinoma epithelial cells (A549) to develop a cell-based electrochemical biosensor for the phloretin measurement [102]. In this study, A549 cells were first encapsulated in alginate and then immobilized onto GCE modified with self-assembled L-cysteine/AuNPs (Figure 5b). Alginate with high stability and biocompatibility was a suitable matrix to carry living A549 cells. On the other hand, L-cysteine provided numerous active sites for binding analytes. The principle of detection was based on safe keeping of antioxidant on A549 cells towards hydrogen peroxide (as a strong oxidizing agent) and direct electron transfer of the oxidative stress reaction. After exposing A549 cells to H_2_O_2_, oxidative harm led to the start of a signaling cascade, resulting in the release of associated proteins. Therefore, the impedance response of A549 cells was evaluated under oxidative stress conditions. In the existence of phloretin antioxidant, the signal intensity of R_et_ decreased in a dose-dependent manner. There was a considerable link between reactive oxygen species (ROS) values and R_et_ values. The response obstacle of the proposed biosensor was linear to phloretin concentrations from 20 μM to 100 μM with the LOD of 1.96 μM. The assay was used for phloretin analysis. The biosensor was stable during 10 days storage at −80 °C.

There are only a few studies on cell-based electrochemical biosensors using microbial cells. For instance, Zhang et al. proposed a biosensor depend on a mutated bacterial laccase WlacD immobilizing onto *Escherichia coli* surface [103]. The recombinant *E. coli* MB275 cells surface-expressing the fusion protein (InaQN)_3_/WlacD were directly adsorbed on a GCE. *E. coli* MB275 showed a stable whole-cell enzymatic activity at pH 2.0–3.0. Several phenolic compounds including catechol, caffeic acid, dopamine, gallic acid, and 2-aminophenol were investigated by the fabricated biosensor with a LOD of 1.0–5.0 µM and linear range of 5.0–500.0 µM.

#### Advantages, Limitations, and the Potential for Practical Applications of Cell-Based Biosensors

Cell-based biosensors have seen considerable evolution in the last ten years due to their distinctive advantages including high sensitivity and selectivity, noninvasiveness, and high biocatalytic activity. However, they are mostly used for toxicological assessments and only a few studies have been conducted on antioxidant analysis using these types of biosensors.

When cells are used as bioreceptor, other biomolecules such as enzymes vital for sensing exist in their native structures and thus exhibit optimal activity and selectivity towards the target molecule [104]. Therefore, due to the more complex physiological structure of the cell compared to the enzyme or nucleic acid, several points should be considered when using them as bioreceptors in antioxidant analysis. In this regard, maintaining the health and the function of the cell during its integration into the biosensor structure is of great importance. New immobilization, conjugation and encapsulation methods are able to preserve cell viability and stability during analysis or storage. As mentioned above, for maintaining cell activity, alginate polymer with outstanding biocompatibility, electrical conductivity, good hydrophilicity, and high mechanical strength is a suitable matrix to encapsulate cells. However, despite cell encapsulation in these studies, cell-based biosensors developed for antioxidants analysis exhibited short-term stability (up to 10 days) during storage than other types of biosensors. In some cases, the biosensor even needed to be stored in certain conditions, such as very low temperatures (–80 °C). Therefore, the development of new methods to improve the stability of this group of biosensors in order to commercialization is of great importance. Another point is the use of recombinant cells with the ability to express specific molecules to expand the sensitivity of the biosensor. For example, when the target molecule is identified by cellular enzymes, the use of recombinant cells with high-throughput enzyme expression can lead to high sensitivity by stabilizing fewer cells, since the number of cells has a positive relationship with the electrode impedance signal. On the other hand, integration of nanomaterials into biosensor either to stabilize the cells or to modify the surface of the electrode can increase sensitivity due to their potential advantages (as mentioned earlier). In this case, the use of nanomaterials with good biocompatibility is desired to maintain cells stability.

#### 3.3. DNA-Based Electrochemical Biosensors

Nucleic acid-based analytical devices have attracted much interest during the last few decades due to their unique inherent features like high stability and particularity, low-price synthesis and smaller size compared to other bioreceptors such as enzyme and antibody [105]. Nucleic acid probes (single-strand DNA (ssDNA), double strand DNA (dsDNA), and purine and pyrimidine bases) have been used as bioreceptors for manufacture of different types of biosensors and for the diagnosis of a variety of analytes including toxins, heavy metals, microbial cells, nucleic acids, hormones, antibiotics etc. [106,107,108,109]. They can be also an excellent bioreceptors for antioxidant analysis based on the fundamental of oxidative damage assessment after exposure to oxidizing agents. In fact, the response of DNA-based electrochemical biosensors for the antioxidant analysis is identical to the effect of antioxidant activity in living cells (in vivo). In DNA-based electrochemical sensors, any shift in the oxidation peak of the DNA bases before and after the interaction with the target molecule will be evaluated. In the presence of antioxidant compounds, they compete with DNA for the hydroxyl radicals, which increase the oxidation signal of DNA determining the antioxidant capacity of sample incidentally [110]. During antioxidant monitoring by DNA-based electrochemical biosensors, the DNA signal is preserved almost unchanged because of the ability of antioxidant molecules in neutralizing the compounds that are causing damage to DNA. As regards DNA oxidative damage is irrevocable; the fabricated electrode can only be utilized once which has its own advantages including high reproducibility, avoiding contamination, and constant sensitivity [111].

The antioxidant activity of chlorogenic acids and coffee was evaluated based on the principle of the dsDNA sensitivity towards OH• radicals [112]. In this electrochemical biosensor, dsDNA probes were immobilized onto SPCE modified with the carboxyl functionalized SWCNTs. The functionalized electrode was incubated (15 min) into a solution containing the optimized concentration of H_2_O_2_ as cleavage agent and DNA damage was evaluated by cyclic voltammetry. An addition of chlorogenic acids and aqueous coffee extracts to cleavage agent remarkably decreased the degree of DNA degradation which was recorded by cyclic voltammetry and expressed as the relative portion of survived DNA (%). Chlorogenic acids content in coffee extract samples were determined by DPV and compared to HPLC method. A good link was seen between the two methods.

In biological systems, DNA damage can occur by hydroxyl radicals generated as a result of the reaction of H_2_O_2_ and metal cations such as Fe^2+^ and Cu^2+^. This process, called the Fenton reaction, is a main feature in the generation of ROS in cellular systems and oxidative harm in tissues leading to mutation and cancer [113]. The generated ROS leads to DNA damage by substitutions in DNA bases or breakage of the DNA strand which can provide a variety of electrochemical responses depending on the kind of harm [114]. Since antioxidant compounds can stop the Fenton reaction, Hashkavayi et al. studied the competency of an *Acanthophora* algae extract with antioxidant properties to inhibit DNA damage inspired by Fenton reactions by an electrochemical biosensor [115]. For the construction of biosensor, a human interleukin-2 (IL-2) gene probe was immobilized on the surface of green synthesized AuNPs-modified carbon screen-printed electrode (SPE). Subsequently, the modified electrode was revealed to the damaging solution containing copper(II) sulfate pentahydrate and H_2_O_2_ in the presence and absence of *Acanthophora* extract for 1 h and investigated by EIS method. In the absence of *Acanthophora* extract, charge transfer resistance (R_ct_) decreased as a result of the Fenton reaction and DNA damage, while amount of harm decreased with enhancing concentration of the extract. Finally, at a concentration of 0.20 mg mL^−1^, the amount of damage was about zero. The defensive impact of *Acanthophora* extract was associated with the presence of phenolic compounds which were measured by distinctive pulse voltammetry approach on the surface of AuNPs-modified SPE. The extract showed inhibitory effect on DPPH and ABTS (2,2’-azino-bis(3-ethylbenzothiazoline-6-sulphonic acid) free radicals. The DNA/AuNPs/SPE biosensor showed good reproducibility and long-lasting stability (40 days at 4 °C). In a similar study, DNA-based biosensor was fabricated by immobilizing dA20 oligonucleotide probe onto carbon-paste electrode (CPE) and oxidative of oligonucleotide was investigated into the Fenton solution (readily made by mixing Fe^2+^, EDTA and H_2_O_2_) in the absence/presence of antioxidants or real samples [116]. After 30 s (reaction time) immersion in Fenton solution, electrochemical quantifications were done by square wave voltammetry (SWV) method. Nine plant infusions were investigated regarding to their polyphenolic compounds and antioxidant activity by the developed biosensor. Among them, green and black tea exhibited higher polyphenolic content and antioxidant capacity. The biosensor fabrication and operation were simple and the reaction time was short.

For electrochemical diagnosis of biophenol oleuropein, a label-free DNA-based electrochemical biosensor was developed by chitosan coating on the surface of CPE [117]. Chitosan as a biocompatible, environmental and non-toxic cationic polymer formed a polyelectrolyte complex with DNA and immobilized DNA on the electrode surface. The oxidation peak of the assembled oleuropein molecules at the modified CPE was measured by using DPV as an analytical signal, because oleuropein is recognized an electroactive species. In fact, oleuropein binding on the surface of modified electrode changed the oxidation signal of DNA, such that the oxidation signal of oleuropein at the DNA-immobilized electrode significantly increased compared to the DNA-free electrode. These observations confirmed the pre-concentration of oleuropein because of the interaction with the DNA layer immobilized on the electrode surface. The biosensor illustrated high sensitivity with a LOD of 0.090 µM and a linear range of 0.30–0.12 µM. Olive leaf extract was used as a food matrix for oleuropein determination by the proposed biosensor.

##### Advantages, Limitations, and the Potential for Practical Applications of DNA-Based Biosensors

Despite the many advantages of DNA-based biosensors due to the nature of DNA, there are still some tips to enhance the performance of the biosensor. DNA immobilization is an essential step for the development of DNA-based biosensor. There are several techniques for DNA immobilization including adsorption, covalent immobilization, and avidin-biotin interactions. Choosing the suitable method affects sensitivity, selectivity and stability of fabricated biosensor. For example, non-specific adsorption can affect detection limit and lead to incorrect negative or positive outcomes. Covalent attachment is the most common DNA immobilization method shows high stability. However, this approach is time-wasting and immobilization yield is almost low. Adsorption through the electrostatic attraction between the negatively charged DNA and the positively charge surface is the simplest technique which does not require any nucleic acid modification. In adsorption technique, ssDNA can immobilize by applying a potential to an electrode. Therefore, the type of technique should be selected based on the desired purpose. In the case of specificity, most of DNA-based biosensors for antioxidant analysis are non-specific because of the detection mechanism is based on the inhibitory effect of antioxidant molecules on oxidative damage of DNA caused by oxidizing agents such as H_2_O_2_ or hydroxyl radicals. Therefore, these types of biosensors are suitable for general analysis of antioxidants or determination of antioxidant activity of a substance. DNA-based biosensors usually show high sensitivity. However, the use functional materials conducting polymers or nanomaterials with suitable electrocatalytic activity can remarkably improve the electrical signal and thus increase the sensitivity of the method. These materials can be used for electrode modification. Moreover, it has been observed that ssDNA is more appropriate for TAC analysis than dsDNA because the bases in ssDNA possess better access to the electrode surface which simplify the oxidation reaction and thus the produced current signal is higher with the enhanced sensitivity. Compared to the cell- and enzyme-based electrochemical biosensors for antioxidant analysis, DNA-based sensors showed a lower reaction time and thus decreased detection time which is suitable for on-site detection and commercialization.

#### 3.4. Other Electrochemical Sensors and Biosensors for Antioxidant Analysis

Laccase and tyrosinase are the most common enzymes utilized for electrochemical antioxidant biosensors. However, xanthine oxidase can be a good option for an enzyme-based electrochemical biosensor. An amperometric biosensor using SPE modified with Prussian Blue (PB) and xanthine oxidase was utilized for the diagnosis of antioxidant activity in Amazonian fruit samples [118]. Xanthine oxidase was immobilized by photo-polymerization into an azide-unit pendant water-soluble photopolymer. The detection principle was based on the online monitoring of the H_2_O_2_ generated during oxidation process of the aqueous hypoxanthine to uric acid in the existence of the xanthine oxidase or by spontaneously dismutation of the superoxide anion radicals. The produced H_2_O_2_ was reduced on the polarized electrode surface, in the presence of Prussian Blue mediator, which is famous for its catalytic impact for the H_2_O_2_ depletion related to the its particular chemistry structure. In the presence of antioxidants, the superoxide anion radicals or H_2_O_2_ were scavenged resulting in a reduction of the cathodic current showing the antioxidant capacity of sample. The suggested biosensor was uncomplicated, low-cost, portable, stable and sensitive with LOD of 2.17 µM and linear range of 1.0–75.0 µM. The approach was applied for the analysis of pure gallic acid and antioxidant capacity of Amazonian fruits samples. Despite many advantages, the biosensor suffered from short-term stability (2 days) during storage at 4 °C.

Proteins are other important macromolecules in cells which are damaged by exposing to oxidizing agents, so they can also be used to develop biosensors for antioxidant analysis. However, no studies have been conducted on protein-based antioxidant biosensors for at least the last five years. In addition to biological components such as DNA, enzyme and cell which have been widely used for antioxidant analysis, a small number of other elements have also been studied as receptor. Molecularly imprinted polymers (MIP) as artificial recognition elements with high affinity and selectivity towards target molecules show a significant potential for detection of different analytes [119]. They can also be utilized for highly sensitive recognition of antioxidants. In this regard, a low-cost, simple, sensitive and selective electrochemical sensor based on MIP and MWCNTs-modified CPE was developed for gallic acid determination in fruit juices [120]. For the surface modification of CPE, MIP was mixed with MWCNT and graphite and used to fill a hole at the end of an electrode body. DPV was used for gallic acid measurement so that with increasing concentration of gallic acid, the peak oxidation increased. At the optimal circumstances, the suggested sensor illustrated a LOD of 47.0 nM with a wide linear range of 0.12 to 380.0 µM. The method was applied to distinguish gallic acid in apple, pineapple, orange juices, and a commercial green tea drink. The MIP-based sensor was highly selective to determine gallic acid in the presence of other interfering compounds. Moreover, a short accumulation time of 14.5 min for gallic acid detection made it an ideal rapid method for on-site detection. However, the sensor showed short-term stability (7 days at 4 °C) which can be improved by changing the method or material used for MIP synthesis. According to these results, MIPs can be promising recognition elements for selective determination of antioxidants.

Nanomaterials with oxidase-like activity, termed as “nanozymes”, and numerous advantages over natural enzymes, can be a good option to construct antioxidant biosensors. Among different nanomaterials with oxidase-like properties, cerium oxide NPs (CeO_2_NPs) or nanoceria particles have gained considerable awareness because of their distinctive catalytic and free radical scavenging features related to the dual reversible oxidation states of cerium Ce^3+^/Ce^4+^ onto the nanoparticle surface [121]. Based on this property, CeO_2_NPs can act as catalyst and imitate the activity of oxidase and peroxidase enzymes. These nanoparticles are inexpensive, stable, insusceptible to denaturation and robust for development of analytical devices. Biomimetic nanoceria was used for the evolution of a disposable single use electrochemical sensor for antioxidants analysis [122]. Nanoceria-modified SPCE catalyzed the oxidation of phenolic mixtures, especially those with *ortho*-dihydroxybenzene functionality, to their corresponding quinones on the surface of the electrode and electrochemical depletion of the produced quinone was measured at −0.1 V vs. the Ag/AgCl electrode. The LOD of sensor for gallic acid, caffeic acid, quercetin and ascorbic acid was estimated to be 1.5, 15.3, 8.6 and 0.4 µM, respectively. The suggested sensor was used for the analysis of antioxidant content in wine samples with high selectivity towards other interfering compounds in wine. Moreover, short-response time (40 s), high stability (months or years at 25 °C), one-step detection, and ease-of-preparation were other advantages of nanoceria-based electrochemical sensor. In another study, AuNPs were used as nanozyme due to their enzyme-like activity, identical to that of natural peroxidases [123]. AuNPs were deposited on the surface of SPE. Using the label-free electrochemical sensor, TAC of plant extracts was determined by monitoring the scavenging capacity of antioxidants present in the plant extracts towards H_2_O_2_. For this purpose, electrical current of H_2_O_2_ was identified in the absence and presence of each plant extract. If the extract contained antioxidant compounds, the electrical current decreased. A good link was found between the results of electrochemical sensor with those obtained by classical chemiluminescence method.

In addition to nanoceria and AuNPs, few other nanomaterials such as PtNPs and Fe_2_O_3_ have recently been studied for oxidase-like activity which can be proposed to fabricate electrochemical sensors for antioxidant assessment.

Metal-organic frameworks (MOFs) are a series of hybrid micro- or nano- crystalline porous materials with uniform structures [124]. They are coordination polymers synthesized via coordinate bonding between inorganic metal ion clusters and organic ligands [125]. Due to their high porosity and tunable physical and chemical features, MOFs have been used in a variety of fields, including medicine, food safety, environmental analysis, drug delivery, etc. In food sector, MOFs have been employed for detection and monitoring of contaminants and other analytes in food products [124]. MOFs exhibit unique properties such as high specific surface areas, defined chemical structures, open metal active sites, defined periodic crystal structures and tunable surface functionalities, which make them fabulous materials with high sensitivity in electrochemical sensing applications. In this regard, several electrochemical sensors based on MOFs have been developed for antioxidants analysis. Li et al. determined ascorbic acid using an electrochemical sensor based on MOFs. As shown in Figure 6a, the sensor was prepared by in-situ growing MOFs (ZIF-65) on the surface of carboxylated CNTs [126]. ZIF-65 (Zeolitic imidazole framework-65), self-assembled from Zn^2+^ and 2-nitroimidazole (2-nIm), is one of representative MOFs. The exposed nitro groups on the frameworks can be utilized as redox active sites.

The fabricated nanohybrid was dropped onto GCEs to fabricate a modified ZIF-65@CNTs electrode (Figure 6b). The designed sensor showed enhanced conductivity due to application of CNTs. Moreover, the porous crystal structures, high water-stability and oxidizing nitro groups of ZIF-65, resulted in high sensory performances of the sensor to ascorbic acid. The LOD and linear range were determined 1.03 µM and 200–2267 µM, respectively.

MOFs have been incorporated with different kinds of nanomaterials to increase the sensitivity of detection methods. In this regards, a non-enzymatic electrochemical sensor for the highly sensitive measurement of catechol was developed based on the layer-by layer modification of the GCE surface with a copper MOF (Cu-MOF), ZnTe nanorods (ZnTe NRs) and AuNPs [127]. As represented in Figure 7, Cu-MOF and ZnTe nanorods were synthesized through solvothermal method. Then, a suspension of Cu-MOF/ZnTe NRs was prepared by mixing the powders of Cu-MOF and ZnTe NRs with an appropriate ratio in de-ionized water. The Cu-MOF/ZnTe NRs/Au NPs/GCE was constructed by a simple layer-by-layer modification protocol. Then, AuNPs were casted on the dry Cu-MOF/ZnTe NRs/GCE. The prepared composite electrode showed an excellent electrocatalytic activity with increased electrochemical response towards the oxidation of catechol, due to the synergistic effect of Cu-MOF/ZnTe NRs and AuNPs. Under optimized conditions, the electrochemical sensor showed a LOD of 16 nM with a wide linear range from 0.25 µM to 300 µM. The developed sensor exhibited enhanced catalytic properties, anti-interference ability, excellent reproducibility and stability, indicating the MOF-based nanocomposite could be used as a promising sensing platform for the measurement of catechol and its derivatives. The developed sensor was successfully used for the determination of catechol and its derivatives in the pharmaceuticals, water and tea samples.

In another study, ZIF-9 was selected as the cobalt-based MOF precursor [128]. The ZIF-9-derived cobalt oxide porous carbon material (Co_3_O_4_@C) was employed as a special substrate to disperse AgNPs. As illustrated in Figure 8, the synthetized Co_3_O_4_@C suspension was dropped onto the surface of GCE. Then, in order to fabricate AgNPs/Co_3_O_4_@C/GCE, the electrode was prepared by electrodeposited in the solution containing AgNO_3_. The modified electrode was used for monitoring of the oxidative stress of living cells and assessment of capacities of scavenging O_2_
^•−^ of food antioxidants at cellular milieu. The electrochemical sensor showed a low LOD of 0.0564 pM. The developed electrochemical sensor showed excellent selectivity and reproducibility.

Representative examples of recent developed electrochemical biosensors for the determination of antioxidants have been listed in Table 3.

## 4. Conclusions

Sensor/biosensor technology has penetrated numerous fields including food analysis, medicine, forensic medicine, drug screening and environmental monitoring. In recent times, much attention has been given to the analysis of antioxidants, as a group of important components of foods, using sensors/biosensors and assays/bioassays due to several advantages of these methods including high sensitivity, quick responses, simplicity of use and ease of miniaturization which make them suitable for on-site diagnosis. The analysis of antioxidants by optical sensors/biosensors based on colorimetric or fluorescence signal provide a simple detection platform. Colorimetric assays are very popular, simple, and convenient and display great value for on-site detection. In this regards, colorimetric paper-based analytical devices are considered as emerging tools due to their simplicity, portability, user-friendliness and cost-effectiveness. Among different kinds of color generating probes, nanomaterials have been extensively utilized in antioxidant sensors because of high surface area, high stability, and high reactivity. The most commonly used nanomaterials in colorimetric assays include AuNPs, AgNPs and CeONPs. Among them, CeONPs provides assays with good precision and higher sensitivity. As listed in Table 1, most of colorimetric assays for antioxidant analysis show low LOD, short detection time and high stability during storage, which make them suitable approaches to enter the market and commercialize. However, a semi-quantitative or even qualitative result is a main limitation of colorimetric assays. This issue can be addressed by designing suitable colorimetric Apps that can be installed on smartphones. On the other hand, fluorescence-based assays with high sensitivity and quantitative results can be easily utilized for antioxidants analysis. However, there are only a few researches on fluorescence sensors/biosensors for antioxidants assessment in food samples. Similar to the colorimetric assays, nanomaterials have been widely employed in fluorescence-based antioxidants assays. In the meantime, QDs have received more attention to develop fluorescence assays.

Electrochemical sensing strategies with high sensitivity and accuracy for antioxidant analysis have been designed using different types of electrodes, transducers and receptors. Enzymes, cells, DNA, and molecularly imprinted polymers (MIPs) have been used as receptors for antioxidants assessment. DNA-based biosensors are the first option for antioxidant analysis because of the direct attack of free radicals on DNA identical to what happens in living cells. Enzyme-based biosensors are easy to fabricate and reusable, however, the low stability of enzymes is an important limitation. Cell-based biosensors require complicated fabrication processes and more detection specificity and stability which are big challenges for commercialization. MIPs are emerging receptors with high stability and low-cost synthesis which have so far rarely been used in the evolution of antioxidant biosensors and deserve further study. In most electrochemical sensors/biosensors, integration of nanomaterial into electrochemical sensor/biosensor resulted in a highly sensitive assay. Several points that should be considered for future works include development of analytical tools with high sensitivity and reliability, long-term stability, portability, and applicability in complex food matrices. Moreover, accurate identification of detection mechanisms in receptors such as cells with complex structures compared to enzymes and DNA is essential. Development of portable readout equipment in all detection strategies can provide a simple and on-site analysis.

## Data Availability

Not applicable.

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
