# Peer review of "An Overview of Optical and Electrochemical Sensors and Biosensors for Analysis of Antioxidants in Food during the Last 5 Years"

_sensors, 2021, doi:10.3390/s21041176_

Round 1
Reviewer 1 Report
Manuscript Number: 1077865
Title: An overview of optical and electrochemical sensors for analysis of antioxidants in food (during the last 5 years)
The manuscript is correctly written. The overall style is verbose, often consisting of a series of references with a sentence of qualitative description. It is necessary to critically compare the described sensors.
Major concerns:
Description of the buildup of different biosensors types is poor. Important characters, like lower limit of detection, response time, stability, slope of calibration are not discussed in details etc.
Minor concerns:
59, Page 3:
Boring?
94-96, Page 5:
Paper based analytical tools are recognized as a practical method just now, that can be used as a recognition platform in food safety
Comment: reformulate a sentence, is not clear
97, Page 5:
Suupy?
100-102, Page 5:
Among these labels, the most frequent one in colorimetric assays 100 for determination of antioxidant capacity is nanoparticles, that are including, gold nanoparticles 101 (AuNPs), silver nanoparticles (AgNPs), cerium oxide nanoparticles (CeNPs, nanoceria), etc.
Comment: reformulate a sentence, is not clear
240, Page 10:
Ascorbic acid
Comment: Ascorbic acid – use lower case letter
252, Page 11:
gallic Acid
Comment: Acid – use lower case letter
268, Page 12:
diphenyl 1 picrylhydrazyl
Comment: use diphenyl-1-picrylhydrazyl
275-277, Page 12:
The proposed sensor was high sensitive than the solution-phase ferric-phenanthroline approach because the membrane concentrated the colored species from a larger volume solution.
Comment: reformulate a sentence, is not clear
369-371, Page 16:
Specific attention has been assigned to approaches with having advantages of sensitivity, quickness, low requirement of sample, required simple and low cost instrumentation, with a goal to rationalize the utilize of research resources
Comment: reformulate a sentence, is not clear
373-376, Page 16:
Nanomaterials have unique optical, electrical, thermal, mechanical, magnetic and biological characteristics that depended to size and can be adjusted by adaptation of the size, shape and the range of accumulation.
Comment: reformulate a sentence, is not clear
511-513, Page 22:
Although fluorescence methods are sensitive, they require identification by fluorometers, which probably not commonly accessible in analytical laboratories.
Comment: reformulate a sentence, is not clear
896-898, Page 36:
Because oleuropein is an electroactive species, the oxidation peak current of the assembled oleuropein molecules at the modified CPE was measured as analytical signal using DPV.
Comment: reformulate a sentence, is not clear
General objection: The entire text should be carefully reviewed and corrected (hardly understandable and confused sentences, etc.).
Check the whole text on the omitted space.
Author Response
The manuscript is correctly written. The overall style is verbose, often consisting of a series of references with a sentence of qualitative description. It is necessary to critically compare the described sensors.
Q1: Description of the buildup of different biosensors types is poor. Important characters, like lower limit of detection, response time, stability, slope of calibration are not discussed in details etc.
Answer 1: Thank you very much for your precision. However, almost all of important characters were mentioned in tables in except of slope of calibration, which wasn’t necessary in articles to be reported. Even, according to the suggestion of one of reviewers, it was better to report them in tables instead of text of manuscript.
Q2: 59, Page 3: Boring?
Answer 2: Thanks for your finesse. It was changed to required complicated procedures (Line 60).
Q3: 94-96, Page 5: Paper based analytical tools are recognized as a practical method just now, that can be used as a recognition platform in food safety
Comment: reformulate a sentence, is not clear
Answer 3: Thanks for your comment. The sentence was changed to “paper based devices are currently known as an effective and alternative method and have been applied as a detection platform in several areas including food safety, environmental monitoring and clinical analysis” (Lines 97-100).
Q4: 97, Page 5: Suupy?
Answer 4: It was changed to supply (Line 101).
Q5: 100-102, Page 5: Among these labels, the most frequent one in colorimetric assays 100 for determination of antioxidant capacity is nanoparticles, that are including, gold nanoparticles 101 (AuNPs), silver nanoparticles (AgNPs), cerium oxide nanoparticles (CeNPs, nanoceria), etc.
Comment: reformulate a sentence, is not clear
Answer 5: It was changed to “noble metal nanoparticles are the most frequent one in colorimetric assays for detection of antioxidant capacity. They include gold nanoparticles (AuNPs), silver nanoparticles (AgNPs), cerium oxide nanoparticles (CeNPs, nanoceria), etc. (Lines 104-106).
Q6: 240, Page 10: Ascorbic acid
Comment: Ascorbic acid – use lower case letter
Answer 6: It was changed to ascorbic acid (Line 245).
Q7: 252, Page 11: gallic Acid
Comment: Acid – use lower case letter
Answer 7: It was changed to gallic acid (Line 257).
Q8: 268, Page 12: diphenyl 1 picrylhydrazyl
Comment: use diphenyl-1-picrylhydrazyl
Answer 8: It was changed to diphenyl-1-picrylhydrazyl (Line 273).
Q9: 275-277, Page 12: The proposed sensor was high sensitive than the solution-phase ferric-phenanthroline approach because the membrane concentrated the colored species from a larger volume solution.
Comment: reformulate a sentence, is not clear
Answer 9: It was changed to “the sensor was more sensitive than the solution-phase method because the membrane concentrated the color from a larger volume solution” (Lines 280-282).
Q10: 369-371, Page 16: Specific attention has been assigned to approaches with having advantages of sensitivity, quickness, low requirement of sample, required simple and low cost instrumentation, with a goal to rationalize the utilize of research resources
Comment: reformulate a sentence, is not clear
Answer 10: It was changed to “particular focus has been given to methods with high sensitivity, quickness, low requirement of sample, required simple and low cost instrumentation, with a purpose to rationalize the use of research resources” (Lines 369-371).
Q11: 373-376, Page 16: Nanomaterials have unique optical, electrical, thermal, mechanical, magnetic and biological characteristics that depended to size and can be adjusted by adaptation of the size, shape and the range of accumulation.
Comment: reformulate a sentence, is not clear
Answer 11: It was changed to “nanomaterials have special thermal, mechanical, optical, electrical, magnetic and biological properties, which are size- dependent and can be tuned by simply adjusting the size, the shape and the extent of agglomeration” (Lines 372-375).
Q12: 511-513, Page 22: Although fluorescence methods are sensitive, they require identification by fluorometers, which probably not commonly accessible in analytical laboratories.
Comment: reformulate a sentence, is not clear
Answer 12: It was changed to “Although fluorescent methods are sensitive, they require detection by fluorometers, which may not be routinely available in analytical laboratories (Lines 513-515).
Q13: 896-898, Page 36: Because oleuropein is an electroactive species, the oxidation peak current of the assembled oleuropein molecules at the modified CPE was measured as analytical signal using DPV.
Comment: reformulate a sentence, is not clear
Answer 13: It was changed to “the oxidation peak of the assembled oleuropein molecules at the modified CPE was measured by using DPV as an analytical signal, because oleuropein is recognized an electroactive species” (Lines 894-896).
Q14: General objection: The entire text should be carefully reviewed and corrected (hardly understandable and confused sentences, etc.).
Check the whole text on the omitted space.
Answer 14: Thank you very much for your attention. The entire text was checked and confused sentences were revised.
Reviewer 2 Report
The authors in the review paper entitled "An overview of optical and electrochemical sensors for analysis of antioxidants in food (during the last 5 years)" have described the progress in different types of optical and
electrochemical biosensors for the analysis of antioxidants in foods. They have compared the different types of sensors in the determination of antioxidants in food samples. It is actually a plus point of this review paper.
However, there are many flaws in the writing style. For example:
- The figures have never been described in the main text. It seems that they have tried to include them forcefully without realizing the importance.
- The ratio of the number of figures as compared to the size of the text in the paper is quite low. It is highly suggested to increase the figures.
- The authors have unnecessarily increased the text size by repeating the sentences while comparing the pros and cons of the different sensors. Instead of text, it is suggested to compare them in a table. That way, the readership can find the required information easily.
- There are many grammatical errors in the review paper. The sentences are not flowing smoothly. It is suggested to review the grammar of the review paper.
Author Response
The authors in the review paper entitled "An overview of optical and electrochemical sensors for analysis of antioxidants in food (during the last 5 years)" have described the progress in different types of optical and
electrochemical biosensors for the analysis of antioxidants in foods. They have compared the different types of sensors in the determination of antioxidants in food samples. It is actually a plus point of this review paper. However, there are many flaws in the writing style. For example:
Q1: The figures have never been described in the main text. It seems that they have tried to include them forcefully without realizing the importance.
Answer 1: Thank you very much for your valuable comment. For Fig.1a, it was added; actually, AuNRs were used as seeds for the epitaxial growth of the Ag, which could be generated from the redox reaction between antioxidants and silver nitrate (Lines 182-184). For Fig. 1b, it was added; the formation of AgNPs caused by gallic acid in a presence of PVA-AgNPs seeds was showed (Lines 200-201). For Fig. 3a, it was added; 3D-printed apparatus with dark chamber contained a strip hole where the paper strip went through. Each spot was processed, one at the time, to reach the UV LED area. Phenolic mixtures susceptible of producing GQDs quenching and those couldn’t produce it were denoted with a yellow and red circle, respectively (Lines 452-455). For other figures, the explanations were enough.
Q2: The ratio of the number of figures as compared to the size of the text in the paper is quite low. It is highly suggested to increase the figures.
Answer 2: Many articles either did not have a schematic figure or did not have a suitable figure. So, We tried to include best figures. However, with respect, new figures were added: Fig. 6 (Lines 1024-1025), Fig. 7 (Lines 1046-1047) and Fig. 8 (Lines 1059-1060).
Q3: The authors have unnecessarily increased the text size by repeating the sentences while comparing the pros and cons of the different sensors. Instead of text, it is suggested to compare them in a table. That way, the readership can find the required information easily.
Answer 3: Thanks for your good comment. Unnecessary pros and cons in some parts of manuscript were deleted and replaced in tables.
Q4: There are many grammatical errors in the review paper. The sentences are not flowing smoothly. It is suggested to review the grammar of the review paper.
Answer 4: The text was revised and grammatical errors were corrected.
Reviewer 3 Report
The authors report the recent done efforts for developing optical and electrochemical sensing platforms for the detection of the antioxidants.
I believe this effort is deserved to be reported by the journal, which may arouse great interest for relevant scientists to review the most recent work under this topic. However, there are some corrections needed and suggestions before acceptance:
- The title of the review is different on the journal website from the title in the file; Is it "sensor" or "biosensor".
- The authors should add a part about the efforts for detection of the antioxidants based on metal-organic-frameworks (MOF), there are several articles that have been published using MOFs.
- In line 59, what the authors meant with the word "boring"?
- In line 368, the authors mentioned that "The development of new biosensors...." while all the described work in this section are not "biosensor" but chemical sensors based on nanomaterials.
- The introduction will need to be revised as it is describing the advantages of the biosensors while the colorimetric section focused on the nanomaterials.
Author Response
The authors report the recent done efforts for developing optical and electrochemical sensing platforms for the detection of the antioxidants.
I believe this effort is deserved to be reported by the journal, which may arouse great interest for relevant scientists to review the most recent work under this topic. However, there are some corrections needed and suggestions before acceptance:
Q1: The title of the review is different on the journal website from the title in the file; Is it "sensor" or "biosensor".
Answer 1: According to the manuscript which include both sensors and biosensors, the title was corrected to the “An overview of optical and electrochemical sensors and biosensors for analysis of antioxidants in food (during the last 5 years)” (Line 1).
Q2: The authors should add a part about the efforts for detection of the antioxidants based on metal-organic-frameworks (MOF), there are several articles that have been published using MOFs.
Answer 2: Thanks a lot for your interesting suggestion. I really enjoyed this suggestion. This part was added to manuscript (Lines 1000-1061).
Q3: In line 59, what the authors meant with the word "boring"?
Answer 3: It was changed to “required complicated procedures” (Line 60).
Q4: In line 368, the authors mentioned that "The development of new biosensors...." while all the described work in this section are not "biosensor" but chemical sensors based on nanomaterials.
Answer 4: It was changed to sensors in this part of manuscript (Line 372). Also, in some parts of manuscript both sensors/biosensors were added.
Q5: The introduction will need to be revised as it is describing the advantages of the biosensors while the colorimetric section focused on the nanomaterials.
Answer 5: Thanks for your precision. It was revised to sensors and also complementary advantages were added (Lines 72-75).
Round 2
Reviewer 2 Report
Thank you for the revision. The revised version looks fine.
Reviewer 3 Report
The authors have addressed all the comments and suggestions.
Thanks!